# Church, State, and Virtue in Roman Catholic Diocese of Brooklyn v. Cuomo (2020)

Edward A. David 

Faculty of Theology and Religion, University of Oxford, Oxford OX2 6GG, UK; edward.david@bfriars.ox.ac.uk

**Abstract:** To curb the spread of COVID-19, houses of worship in the State of New York were legally required to limit attendance at religious ceremonies. Two religious communities—the Roman Catholic Diocese of Brooklyn and the Orthodox Jewish organization, Agudath Israel of America—asked the U.S. Supreme Court to intervene. This article provides a theological interpretation of the Court's decision to grant these communities injunctive relief, thus freeing them from the State's restrictions on religious attendance. Drawing upon the Catholic tradition, and especially the thought of Saint Thomas Aquinas, the article offers a sustained virtue-based analysis of the Court opinion and of the relationship between church and state more generally.

**Keywords:** religious exemptions; Aquinas; Eucharist; virtue ethics; COVID-19

## 1. Theology in the *Brooklyn* Court

On 7th March 2020, Andrew Cuomo, the then Governor of New York, declared a disaster emergency for the entire state in response to the growing threat of COVID-19 (Executive Order, no. 202). Later that year, on 6th October, Cuomo approved "enhanced public health restrictions" to curb the spread of infection (Executive Order, no. 202.68).

The restrictions—which applied to a range of organizations, including houses of worship—varied in stringency according to zones. In red zones (areas with the highest level of "cluster-based cases of COVID-19"), houses of worship were required to cap attendance at religious ceremonies to "25% of maximum occupancy or 10 people, whichever [was] fewer." In orange zones (areas of "moderate severity"), houses of worship had a capacity limit of 33% or 25 people. Finally, in yellow zones ("precautionary" areas), houses of worship were subject to a capacity limit of only 50% (Executive Order, no. 202.68).

These restrictions were viewed as especially severe by two religious communities—the Roman Catholic Diocese of Brooklyn and the Orthodox Jewish organization, Agudath Israel of America. Both sought legal remedy in the form of injunctive relief from the United States Supreme Court. At issue, the communities argued, was their right to religious liberty as protected by: (i) The free exercise clause of the First Amendment; and (ii) The Court's "minimum requirement of neutrality to religion" (*Church of Lukumi Babalu Aye, Inc. v. Hialeah*, 508 U.S. 520, 533 [1993]), which prohibits the government from treating houses of worship more harshly than their secular counterparts. According to Agudath Israel, "the Governor specifically targeted the Orthodox Jewish community . . . [by] gerrymandering the boundaries of the red and orange zones to ensure that heavily Orthodox areas were included", and with the Diocese, they maintained that "the regulations treat[ed] houses of worship much more harshly than comparable secular facilities" (*Roman Catholic Diocese of Brooklyn v. Andrew M. Cuomo, Governor of New York*, 141 S. Ct. 63, 66 [2020]).[1] Convinced that the communities' First Amendment claims would prevail (that is, if the disputes were fully litigated in court), the Justices granted the communities immediate relief. The Governor was, thus, barred from enforcing the most severe of the attendance restrictions—that is, the 10- and 25-person occupancy limits in particular.

The Court's per curiam, i.e., anonymously authored, opinion features two statements of interest. First, the opinion states that the Governor's restrictions "would lead to irreparable injury" for the religious communities and, second, that a relaxation of the restrictions "would not harm the public interest" (*Brooklyn*, 141 S. Ct. at 66). These statements constitute a legal conclusion that may be reasonably disputed (indeed, it was disputed by Chief Justice Roberts, as well as Justices Breyer, Kagan, and Sotomayor in dissenting opinions). However, while a technical legal analysis is welcome, it will be helpful, too, to interrogate the Court's opinion from a different perspective altogether—that is, from a theological perspective of at least one of the religious communities affected.

The importance of theological interrogation can be discerned in the Court's one-paragraph discussion of irreparable harm. The Court begins by claiming that restrictions on religious attendance will have serious legal consequences, namely, "the loss of First Amendment freedoms", which "unquestionably constitutes irreparable injury" (*Brooklyn*, 141 S. Ct. at 67). This legal point is then complemented with a gesture toward moral and theological reasoning. The opinion reads:

> If only 10 people are admitted to each service, the great majority of those who wish to attend Mass on Sunday or services in a synagogue on Shabbat will be barred. And while those who are shut out may in some instances be able to watch services on television, such remote viewing is not the same as personal attendance. Catholics who watch a Mass at home cannot receive communion, and there are important religious traditions in the Orthodox Jewish faith that require personal attendance (*Brooklyn*, 141 S. Ct. at 67–68).

Here, the Court links legal injury ("the loss of First Amendment freedoms") with moral and theological harms (being "barred" from "important religious traditions"). The connection, however, is left theologically undeveloped, as the paragraph abruptly ends and the discussion gives way to an analysis of public interest. It is reasonable, of course, for courts of a (secular) liberal democracy to stop short of theological explication (Stout 2004). Nevertheless, since court opinions are meant to justify to the public the decisions made (Taylor 2019), it would be helpful to know more about the religious beliefs and practices at stake. With the Court unable to furnish this understanding, theologians may step in to fill the gap.

In what follows, therefore, I offer a Catholic perspective on the *Brooklyn* decision, drawing theological attention to the Court's remarks concerning "irreparable injury" and "public interest" (I leave an Orthodox Jewish analysis, representing *Brooklyn*'s additional applicants, to relevant experts.) My discussion will address three areas of theological significance: first, the relationship between church and state, which operates in the background of any U.S. discussion on free exercise; second, the importance of religious attendance, which, when interfered with, can cause "irreparable injury" of eternal significance; third, our "public interest" in worship, which pertains to the virtue-building effects of religious attendance, as well as to the moral role of the (secular) state. Finally, the discussion will consider the interaction between two virtues in particular, justice and wisdom, which help interpret the *Brooklyn* dissent.

The purpose, and contribution, of this discussion is to substantiate the various moral and theological claims that are implied—or otherwise said but left undeveloped—in the Court's opinion. This exercise, I hope, will be useful for theologians who wish to reflect upon the ethics of religious-attendance restrictions, though without wanting to sift through technicalities of jurisprudence. It will be useful, too, for legal scholars who wish to reflect theologically upon religious exemptions, an area of law that is likely to receive further treatment from the U.S. Supreme Court (Girgis 2022).

A notable aspect of my discussion lies in its sustained virtue-based analysis of church and state. Most scholars of law and religion emphasize the jurisdictional claims of each type of association (Smith 2014, 2016; Laborde 2017). My reflection focuses upon the aretaic instead, shining a light on the distinct ends of church and state, as well as the virtues that are implicated in the execution of their responsibilities. To draw out these points, my

discussion references the Catholic tradition more widely. However, it relies heavily upon the medieval theologian Saint Thomas Aquinas (d. 1274). Already, Aquinas's influence is keenly felt in theological reflections on law (Maritain 1944, 1945; Finnis 1998; Gilson 2010; VanDrunen 2014). Nevertheless, it remains a worthwhile endeavor to bring Aquinas to bear upon a single legal dispute (e.g., see David 2020; Moats 2021), primarily for the conceptual clarity he brings to issues of enduring importance. The *Brooklyn* case is replete with such matters.

## 2. The Different Ends of Church and State

I begin our discussion with a brief reflection on church and state. The distinction between both realms, or types of association, is an ancient idea that has been traced to Pope Gelasius I (d. 496), Saint Augustine (d. 430) and, further back, to the New and Old Testaments (Smith 2014; Wilken 2021; Green 2022). Central to this distinction is a question concerning moral and religious responsibilities. For believers, their responsibilities are neatly captured by Christ's admonition to "render to Caesar the things that are Caesar's, and to God the things that are God's" (Mk 12:17). For the church and state themselves, the responsibilities owed to their members are not so pithily stated.

On this front, Aquinas offers conceptual clarity. Following the Aristotelianism of his age (Kantorowicz 1957; Finnis 1998), Aquinas defines each type of association according to their proper ends, which point to their distinct responsibilities. Beginning with the political community or state, its end is the "common good" of its citizens (*Summa Theologiae* I-II q. 95, a. 4). To maintain the common good, a state may guide, or even coerce, the external acts of its citizens: "[f]or the end of human law," Aquinas says, "is the temporal tranquility of the state, which end law effects by directing external actions" (*ST* I-II q. 98, a. 1). Law is, thus, an important instrument of the state, and so the making, enforcement, and adjudication of laws constitute some of its most important responsibilities.

The church, in contrast, has an end of an entirely different sort. Its end is "everlasting happiness" (*ST* I-II q. 98, a. 1), a state of eternal beatitude or "union with God" wherein persons contemplate and forever live in the "divine essence" (*ST* I-II q. 3, a. 8). To help members reach this end, the church has a responsibility to "preach . . . the Gospel of Christ" (*ST* I-II q. 106, a. 4, ad. 4) and to provide the sacraments instituted by him (*ST* III q. 61, a. 1). This is a grave responsibility. However, in exercising this task, the church cannot externally coerce members into eternal life. A free response of faith is required, and since this action is internal, it must be freely given. As Aquinas notes, "even . . . the Gospel would kill, unless there were the inward presence of . . . faith" (*ST* I-II q. 106, a. 2).

In a well-ordered society, individuals would belong to both types of association and so be subject to the responsibilities and authority of each (Maritain 1944; Finnis 1998; Gilson 2010). The order, or orderliness, of this arrangement may be succinctly described: the state coerces externally, while the church encourages internally. Christians, thus, enjoy a dual citizenship. They are members of an earthly city, the temporal political community, and the city of God, which is prefigured by a pilgrim church (Saint Augustine 1998). At its best, the relationship between both associations is marked by a peaceful interdependence—with the state, for instance, looking to the church for help in civic formation (Hordern 2013), and the church relying upon the state for certain practicalities (Cavanaugh 2014).

Such dependence, notwithstanding, the tension between church and state cannot be overlooked. As Pope Benedict XVI suggests, this tension is important because it pertains to freedom itself. "Each of these communities [i.e., church and state] has a limited radius of activity," he explains, "and keeping their mutual relationship in balance is the basis for freedom" (Pope Benedict XVI 2008, p. 156). What freedom? In the American context, and in the *Brooklyn* dispute especially, we can point to "First Amendment freedoms," the loss of which may "unquestionably constitute[] irreparable injury" (*Brooklyn*, 141 S. Ct. at 67).

### 3. Irreparable Injury and Worship's Saving Effects

Having outlined a basic understanding of church and state (there are many understandings, of course: see Section 5 below), I turn, next, to the Court's discussion of "irreparable injury." As noted above, the Court majority suggests that "First Amendment freedoms" are so important that, if lost, "irreparable injury" may follow (*Brooklyn*, 141 S. Ct. at 67). This legal claim is then complemented with a theological point of fact: "Catholics who watch a Mass at home cannot receive communion," the justices note (*Brooklyn*, 141 S. Ct. at 68). However, one may reasonably ask why this latter point even matters. A legal freedom of worship is clearly implicated. However, the Court majority—in pointing to the specific example of communion—suggests that something deeper is at stake. "Irreparable injury," in other words, must have supernatural significance, as well as legal significance. Two reflections may help illuminate this point.

First, we note the relationship between in-person worship and eternal salvation. Communion—or the sacrament of the Eucharist—is a central form of worship in many Christian traditions (Wandel 2006). However, for Roman Catholics especially, it entails a supernatural reality of the deepest religious significance: an encounter with and reception of the body and blood of Jesus Christ. More than a metaphor, the Eucharist literally is the body of Christ (*Catechism of the Catholic Church*, para. 1365–66), the *corpus verum* or true physical body (De Lubac 2006) that was sacrificed on the cross and "poured out for many for the forgiveness of sins" (Mt 16:28). The Council of Trent, in 1562, explains this significance in the following way:

> [Christ], our Lord and God, was once and for all to offer himself to God the Father by his death on the altar of the cross, to accomplish there an everlasting redemption. But because his priesthood was not to end with his death, at the Last Supper on the night when he was betrayed, [he wanted] to leave to his beloved spouse the Church a visible sacrifice (as the nature of man demands) by which the bloody sacrifice which he was to accomplish . . . would be re-presented, its memory perpetuated until the end of the world, and its salutary power be applied to the forgiveness of the sins we daily commit (quoted in *CCC* para. 1366; internal quotations removed).

As the Council suggests, the forgiveness of sins is part and parcel of a believer's sanctification, which, in turn, is linked to eternal salvation. Catholic tradition, in fact, holds that the physical consumption of the Eucharist is necessary for salvation: "Truly, I say to you, unless you eat the flesh of the Son of man and drink his blood, you have no life in you" (Jn 6:53, quoted in *CCC* para. 1384). Communion, thus, entails a decidedly physical form of salvation, which includes a bodily resurrection at the end of time (*CCC* para. 366, 1384). Certainly, interference with this could amount to "irreparable injury" (*Brooklyn*, 141 S. Ct. at 66).

Building upon this conclusion, we next consider a Eucharistic dimension of church and state. Contemporary scholars of religious liberty often discuss an ancient (legal) doctrine known as libertas ecclesiae, which, in one formulation, asserts a jurisdictional claim of the church, which covers all activities of religious nature—from worship and preaching, to the hiring and retention of church ministers (Moreland 2008; Smith 2011; Schragger and Schwartzman 2013; Garnett 2016; Horwitz and Tebbe 2016). State intervention is unjustified within this jurisdiction, the doctrine claims, due to, in part, a lack of state competency; only the church is qualified to make theological judgements on core Christian beliefs and practices; the state must, therefore, show deference to ecclesiastical decisions (Smith 2016; cf., Weinberger 2022).

This doctrine is accompanied by an interesting theological history that most legal scholars overlook. Outlined by the Jesuit theologian Henri de Lubac, the history involves the development of a medieval tradition, which holds that the church *is* the mystical body of Christ. This means that the church not only is the site of the Eucharistic sacrifice of Christ's body but is also Christ's body in corporate form. As with the Eucharist, no mere metaphor is here deployed: Christ *is* the supernatural head of the church, and Christians *are*

the different parts of Christ's mystical body (1 Cor 6:15: "Do you not know that your bodies are members of Christ?"; see also Harding and Dawes 2009, s.v., "Body"). Stressing this point philosophically, the Benedictine theologian Guy Mansini argues that the (Catholic) church enjoys a unique corporate existence. As a mystical body, it "subsists" in itself, which is to say that there is a

> unique and complete capacity to act [that] is located only in the Catholic church; it makes of her an agent in a unique sense as compared with all other churches and ecclesial communities. This unique agency, moreover, connotes existence as of a hypostasis or individual—subsistence in the philosophical sense (Mansini 2017, p. 52).

It comes as no surprise that legal scholars would, and do, eschew this theological point. Corporate persons, after all, are mere fictions of law (Nelson 2013; Orts 2015; Gindis 2016). Nevertheless, the corpus mysticum doctrine remains relevant for the *Brooklyn* opinion. Consider the following point made by de Lubac: when communicants are "[n]ourished by the body and blood of the Saviour, . . . [it is Christ] who truly makes them into one single body . . . Literally speaking, [then,] . . . the Eucharist *makes* the Church" (De Lubac 2006, pp. 87–88, emphasis added).

Taking seriously these words, one can see how restrictions on religious attendance could isolate believers from God, thus collapsing a vertical relationship between God and the individual; they could also isolate believers from the mystical body of the church, thus breaking a horizontal relationship (a communion) between worshipping members. Legally speaking, then, restrictions on religious attendance can have a material effect upon the collective existence of the church itself—after all, "the Eucharist [literally] makes the church" (De Lubac 2006, p. 88; *CCC* para. 1396). Individual rights to free exercise are here implicated, but so too are freedoms pertaining to religious association. The corpus mysticum doctrine gives these horizontal freedoms a unique theological weight. Indeed, in a very particular sense, there simply is no (experience of) religious association without Eucharistic communion. With a sacred bond broken, "irreparable injury" is an apt term (*Brooklyn*, 141 S. Ct. at 66).

### 4. Public Interest and Moral Virtue

We next consider the *Brooklyn* Court's discussion of "public interest" (*Brooklyn*, 141 S. Ct. at 68). Found in the opinion's final section, public interest is described by the Court with reference to COVID-19:

> The State [of New York] has not claimed that attendance at the applicants' services [i.e., religious ceremonies] has resulted in the spread of the disease. And the State has not shown that public health would be imperiled if less restrictive measures were imposed (*Brooklyn*, 141 S. Ct. at 68).

In other words, the Court acknowledges that there are grave public interests in *not* spreading the disease and in *not* imperiling the health of the general public. "Stemming the spread of COVID-19 is unquestionably a compelling interest," the Court says (*Brooklyn*, 141 S. Ct. at 67).

The concept of "interest" has an intriguing intellectual history, having been associated with "destructive passions (the desire for riches, glory and domination)" by Thomas Hobbes (d. 1679), as well as national and economic interests (including those of "individuals and groups within the nation") in seventeenth-century England (Backhouse 2002, pp. 73–74). The latter, more general understanding still holds today (though, unfortunately, the former "destructive" sense may hold as well). However, rather than focus on the Court's balancing of such interests (instead see Urbina Molfino 2017; Girgis 2022), I wish to draw attention to one of the *Brooklyn* Court's underlying assumptions: the idea that the state should play a significant role in our moral reasoning and life lived together. This much has been implied in Section 2 above. Yet, more can be said about the issue, particularly with regard to virtue and its place within a church–state relationship. Once more, we return to Aquinas.

Aquinas, we recall, posits different ends for the church and state: the latter aims toward the "common good" (*ST* I-II q. 95, a. 4); the former, "everlasting happiness" (*ST* I-II q. 98, a. 1). In posing this distinction, Aquinas departs from a set of Aristotelian ideas, which hold that politics (i.e., the science of the state) is the "master art" (*Nicomachean Ethics*, bk. I, ch. 2) and that it (or the state) makes "citizens to be of a certain character, namely good and capable of noble acts" (*NE* bk. I, ch. 9). Turning these ideas on their head, Aquinas proposes that the church—and not the state—is the most perfect moral community. He makes this claim through a comparison of human and divine law:

> Now human law is ordained for one kind of community [the state], and the divine law for another kind [the church] . . . Wherefore human law makes precepts only about [external] acts of justice . . . , [whose proper function consists in directing the human community,] divine law proposes precepts about all those matters whereby men are well-ordered in their relations to God . . . [This affects] the acts of all the virtues (*ST* I-II q. 100, a. 2).

In the excerpt above, it may be tempting to discern an antagonism between justice and the divine law, and thus, to conclude that the state—and not the church—is concerned with justice. However, Aquinas elsewhere describes "religion" as a form justice, the virtue that gives to others (both God and human persons) their due (*ST* II-II qq. 81–89). Hence, the church—being a community ordered *by* religion or justice *to* the keeping of divine law—is not a de facto enemy of the state. Instead, the church seeks to elevate, even orientate, justice and all the virtues toward the human person's ultimate end: "union with God" (*ST* I-II q. 3, a. 8). All of this suggests that for Aquinas, the church is the most complete moral community, and that the church should have responsibility for developing its members' perfect virtue—this being an internally suasive, not externally coercive, task (Finnis 1998; Budziszewski 2014).

Some of the actions that constitute this task have already been mentioned: they include "preaching . . . the Gospel of Christ" (*ST* I-II q. 106, a. 4, ad. 4) and providing the sacraments instituted by him (*ST* III q. 61, a. 1). We can now add that it is the Mass—the church's "public worship," (*CCC* para. 1199) offered "for all men" (*CCC* para. 1368)—that encompasses both types of actions. It is the Mass, therefore, that serves as a principal means of developing virtue.

The theological reasons behind this claim are many. For one, as the celebration of the Eucharist, the Mass is where believers give due worship to God, expressing "faith in the real presence of Christ under the species of bread and wine" (*CCC* para. 1378). Here, they cultivate the virtue of religion. Second, believers must prepare for the "worthy reception" of the Eucharist (*CCC* para. 1387), recognizing their own faults, forgiving others of theirs (*CCC* para. 1385, 1393), and thus, developing the virtue of humility. Moreover, the worthy reception of the Eucharist involves countless moral and spiritual fruits, including a strengthening of the virtue of charity, an effect of Eucharistic devotion to God (*CCC* para. 1394), renewed commitment to the poor, among whom Christ is counted (*CCC* para. 1397), and a desire for Christian unity, so that all may enjoy "common participation in the table of the Lord" (*CCC* para. 1398). Aquinas describes these and other fruits in a playful way: "it is the soul that is spiritually nourished through the power of this sacrament, . . . . 'Eat, O friends, and drink, and be inebriated, my dearly beloved'" (*ST* III q. 79, a. 1, ad. 2, quoting Song of Solomon 5:1).

In addition to this theological backdrop, virtue formation through the Mass can be understood psychologically. Briefly, we might mention the admiration of Christ and the saints, with their stories experienced through the hearing of scripture and their admirable traits imitated by hearers in the world (Zagzebski 2004, 2017). We might also point to the numerous benefits of attending Mass and other religious services. As Harvard's Tyler VanderWeele (2017) reports, religious attendance is associated with improved mental health, better social relationships, and measurable gains in virtue, and longitudinal evidence shows that those who attend religious services are more generous and civically engaged than those who do not.

So, through its various actions, especially through the Mass, the church is understood to be the most complete moral community. However, now we must ask again about the state, that *other* moral community, which concerns itself with acts of justice (*ST* I-II q. 100, a. 2). What role might it play in the church's deeper, more personal development of citizens' character? To answer this question, I highlight possible moral positions that the state might take toward attendance restrictions.

First, we consider the Governor's position. At worst, Cuomo's restrictions on religious attendance could be seen as embodying state hostility toward religion. In a concurring opinion, Justice Gorsuch offers an acerbic summary of this view. While even "the largest cathedrals and synagogues" are subject to severe restrictions, Gorsuch writes,

> the Governor has chosen to impose *no* capacity restrictions on certain businesses he considers "essential." And it turns out the businesses the Governor considers essential include hardware stores, acupuncturists, and liquor stores . . . So, at least according to the Governor, it may be unsafe to go to church, but it is always fine to pick up another bottle of wine . . . Who knew public health would so perfectly align with secular convenience? (*Brooklyn*, 141 S. Ct. at 69, emphasis in original).

Focusing on the substance of Gorsuch's remarks, one can discern a moral message in the Governor's use of the term "essential": some organizations and activities are important, others are not. Churches and their worship are not (see discussion in *Brooklyn*, 141 S. Ct. at 69). Here, we might say that the state takes a negative position, albeit indirectly, toward the church's worship-based moral pedagogy.

A second moral position to consider is that of the Court majority: "there is no reason why [the Diocese and Agudath Israel] should bear the risk of suffering further irreparable harm," the Court concludes (*Brooklyn*, 141 S. Ct. at 68). Although stated in defensive legal terms, the Court's position lends itself to positive elaboration, both morally and theologically speaking, as this article has endeavored to show. Again, it may not be the Court's place to explicate fine points of theological doctrine or to adopt the thickest of normative language (Rawls 1999; Laborde 2017). However, if such restraint is a virtue of a liberal democratic state, then it can be appreciated as truly virtuous in light of the church's responsibilities toward moral formation (Green 2010; Finnis 2011; West 2017). In other words, to respect individuals, communities, and their rights, perhaps the state *should* refrain from interfering with the moral mission(s) of houses of worship. Undue attendance restrictions, as well as certain forms of moral pontificating from public officials, would amount to unwelcome, even vicious, interventions. Certainly, there is a "public interest" in barring *these* (*Brooklyn*, 141 S. Ct. at 68).

## 5. Wisdom through Disagreement

My reflection thus far has offered a Catholic perspective on "irreparable injury" and "public interest," and may be read as a theological support for the *Brooklyn* decision. However, it is important to recognize that reasonable disagreement can exist over the scope of restrictions on religious attendance. As Chief Justice Roberts notes, one may "simply view the matter differently after careful . . . analysis" (*Brooklyn*, 141 S. Ct. at 75). Such disagreement, moreover, can operate within the domain of virtue, and, indeed, it is in disagreement that virtues are most needed (Grossmann 2017; Vogler 2020). Therefore, to conclude this reflection, we would do well to address yet another moral position that may be read into the Court's reasoning. This is the view that the Governor's restrictions were morally just, being consonant with not only the virtue of religion but also the virtue of practical wisdom. One can read this position in the Court's dissent.

Consider the following argument made by Justice Sotomayor: "state officials seeking to control the spread of COVID-19 . . . may restrict attendance at houses of worship so long as comparable secular institutions face restrictions that are at least equally as strict." In fact, notes Sotomayor, "New York applies similar or more severe restrictions . . . to comparable secular gatherings." These include "lectures, concerts, movie showings, spectator sports,

and theatrical performances, [events] where large groups of people gather in close proximity for extended periods of time" (*Brooklyn*, 141 S. Ct. at 79).

This argument clearly appeals to equality. Therefore, we may ask whether the attendance restrictions follow what the virtue of justice—which is commonly associated with equality—might reasonably demand. Again, Aquinas provides food for thought.

Justice involves the establishment of "equality in our relations with others," writes Aquinas, and this is achieved "by doing good, i.e., by rendering to another [their] due," and "by inflicting no injury upon [our] neighbor" (*ST* II-II, q. 79, a. 1). If "secular gatherings" (universities, concert venues, movie theatres, etc.) justly abide by attendance restrictions to "control the spread of COVID-19" and, thereby, refrain from "exacerbat[ing] the Nation's suffering," then the virtue of justice may expect the same behavior from "comparable [religious] institutions" (*Brooklyn*, 141 S. Ct. at 79). After all, says Aquinas, "it belongs to ... justice to do good in relation to [our] community" (*ST* II-II, q. 79, a. 1), and "all those who are included in that community," from individuals to religious associations, fall within justice's purview (*ST* II-II, q. 58, a. 5).

To this argument, a theological objection may be raised. As discussed in Section 4 above, Aquinas considers religion to be a type of justice, specified by its unique subject—i.e., God—who "infinitely surpasses all things", and thus, deserves a "special honor due" (*ST* II-II, q. 81, a. 4). What could this special honor be? "[S]acrifice, adoration, and the like," says Aquinas (*ST* II-II, q. 81, a. 4). In other words, the "proper and immediate acts" of religion (*ST* II-II, q. 81, a. 1, ad. 1) or, more plainly stated, attendance at public worship. Given this understanding, it may be argued that religious believers *must* attend services so as to give God a "special honor due"; therefore, religious gatherings—in contrast to secular gatherings—*must not* be subject to attendance restrictions. This unequal proposition, the objection holds, is justified by what the virtue of religion commands.

What are we to make of this theological objection? For one, the objection may sit comfortably in the minds of certain legal theorists—for example, with scholars associated with Catholic integralism (Jones 2017; Milbank 2022; Vermeule 2022). However, one challenge that would need to be addressed pertains to religion's second type of act, namely virtuous actions that religion commands for "the honor of God." Such actions, Aquinas notes, include the merciful action of "visit[ing] the fatherless and widows in their tribulation" and the temperate action of "keep[ing] oneself unspotted from this world" (*ST* II-II, q. 81, a. 1, ad. 1). Both actions are commanded by the virtue of religion, and thus, strengthen the general aim of justice, which entails "rendering to another [their] due" (*ST* II-II, q. 79, a. 1). In light of such actions, is it not plausible that religion might, in some very specific circumstance, command that restrictions be placed on religious attendance? Additionally, in that circumstance, might religion invite us to view such restrictions as acts of virtue, which are commanded not only for the good of others but also to honor God?

These questions deserve answers. However, how might we navigate what the virtue of religion specifically demands? For guidance, we turn to the virtue of practical wisdom, a virtue that can be seen in the *Brooklyn* dissent, as well as in the Court majority.

Practical wisdom (or prudence) is often referred to as a master virtue (Peterson and Seligman 2004; Kristjánsson et al. 2021), which helps individuals to deliberate well about the "sorts of thing conducive to the good life in general" (*NE* bk. VI, ch. 5). As Aquinas notes, it helps individuals reason well about things to be done, improving the "appl[ication of] right reason to action" (*ST* II-II, q. 47, a. 4). Additionally, it helps apply universal principles to the particularities of practical matters (*ST* II-II, q. 47, aa. 3 and 6). Prudence, in other words, has an eye toward morally relevant details, and it ensures that such details are factored into our deliberations over difficult moral issues. Decisions over religious-attendance restrictions are easily among the most challenging.

In her dissent, Justice Sotomayor suggests that the Court majority has overlooked morally significant details. Singling out Justice Gorsuch, she writes:

> But Justice Gorsuch does not even try to square his examples [of secular activities that he thinks might pose similar risks as religious gatherings] with the conditions

medical experts tell us facilitate the spread of COVID-19: large groups of people gathering, speaking, and singing in close proximity indoors for extended periods of time. Unlike religious services, which have every one of th[ose] risk factors, bike repair shops and liquor stores generally do not feature customers gathering inside to sing and speak together for an hour or more at a time (*Brooklyn*, 141 S. Ct. at 79).

In a separate dissent, Justice Breyer echoes the same point: "members of the scientific and medical communities tell us that the virus is transmitted . . . when a . . . group of people talk, sing, cough, or breath near each other . . . for prolonged periods of time, particularly indoors or in other enclosed spaces" (*Brooklyn*, 141 S. Ct. at 78). Many religious services provide these exact conditions. Sotomayor and Breyer, thus, offer a morally relevant detail, one that is particularly compelling given the fact that COVID-19 had, at that point, caused "more than 250,000 deaths nationwide . . . with 16,000 [of those deaths] in New York City alone" (*Brooklyn*, 141 S. Ct. at 77). Attentive to such details, the dissenting Justices exhibit moral prudence.

Or do they? Justice Gorsuch may object, specifically to the claim that he did not try to square his arguments with medical advice. In fact, he does consider the measures that houses of worship had taken to minimize the risk of infection; they are practicing "social distancing, wearing masks, leaving doors and windows open, forgoing singing, and [are] disinfecting spaces between services," he writes (*Brooklyn*, 141 S. Ct. at 69). These are measures that, as the Court majority notes, "have complied with all public health guidance." They also entail "additional precautionary measures," going above and beyond what the state required (*Brooklyn*, 141 S. Ct. at 65). Perhaps here we find yet another morally relevant detail, one highlighted by the Court majority, that houses of worship can reduce the risk of transmission by simply altering, not giving up on, their religious services. Once more, with further particulars noted, we see prudence hard at work.

Knowledge of particulars is an integral part of the virtue of prudence (*ST* II-II, q. 47, aa. 3 and 6). Without it, prudence is unable to discern wisely the right course of action. With it, prudence has a fighting chance to navigate the claims of religion, which commands both public worship (given solely to God) and other acts of virtue (which benefit our neighbors). It is not my aim to decide whether prudence rests squarely with the dissent or the majority of the *Brooklyn* Court. Sufficient, instead, is the suggestion that the Governor's restrictions may have been morally just, given particular circumstances.[2] How we know otherwise can only be the result of engaging virtuously with others' opinions and experiences, giving their ideas (including theological beliefs) a just hearing, and, through prudence, being attentive to the details of their perspectives. This lesson holds for opposing sides of the Supreme Court. It also holds for church and state, especially when at odds over their moral—and theological—duties.

**Funding:** This research received no external funding.

**Institutional Review Board Statement:** Not applicable.

**Informed Consent Statement:** Not applicable.

**Data Availability Statement:** Not applicable.

**Conflicts of Interest:** The author declares no conflict of interest.

## Notes

[1] For ease of reading, all internal quotations and citations have been omitted from legal excerpts used in this article.

[2] Similar circumstances have been identified in plagues throughout history (Plüss 2020). Notably, written evidence shows that churches often supported (secular) political authorities in their decisions to interfere with religious ceremonies, so as to stop the spread of contagion (Slack 2020). The differences between then and now are myriad, of course. However, perhaps these historic events and experiences have something to offer for moral reasoning in the future.

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
