# Peer review of "Church, State, and Virtue in Roman Catholic Diocese of Brooklyn v. Cuomo (2020)"

_religions, doi:10.3390/rel14020239_

Round 1
Reviewer 1 Report
I really enjoyed reading this case comment. I thought the theological analysis was particularly pertinent to the analysis of this case. The analysis was well argued and provided a clear and coherent journey for the reader. Although not for this case comment, further study might include a comparison with, for example, the theology of right of Rowan Williams, which takes a similar stance (we do human right at the point when we are in communion with other Christians) and juxtaposed with the theory of common grace found in the reformed tradition (Kuyper et al). This would be beyond the current study which has an appropriate focus on Roman Catholic theology and provides a clear and appropriate analysis of a single legal case.
Three points:
1. Ensure that a non-US audience can understand the first section where you explore the case itself. Using words such as 'enjoined' will be difficult for a non-lawyer and a non-US lawyer to understand.
2. Ensure a brief explanation of the legal provisions follow the explanation of the facts. i.e. what did the claimants and defendants have to prove on the basis of the legal provisions (explain what they say). This helps the non-specialist understand the facts and law.
3. line 44 introduces the fact that one of the religious communities was a Jewish umbrella organisation. It would be worth introducing this earlier when you are explaining the facts. Also explaining earlier (perhaps at the same point) that you will be focussing on the RC aspects. Otherwise the non-US or non-Catholic reader might assume the judgment is about two RC churches.
Thank you for an excellent analysis. I really enjoyed reading this.
Author Response
I really enjoyed reading this case comment. I thought the theological analysis was particularly pertinent to the analysis of this case. The analysis was well argued and provided a clear and coherent journey for the reader.
Although not for this case comment, further study might include a comparison with, for example, the theology of right of Rowan Williams, which takes a similar stance (we do human right at the point when we are in communion with other Christians) and juxtaposed with the theory of common grace found in the reformed tradition (Kuyper et al). This would be beyond the current study which has an appropriate focus on Roman Catholic theology and provides a clear and appropriate analysis of a single legal case.
Thank you for this suggestion. Williams' theology of right, along with Kuyper's sphere sovereignty, would make for a rich theological analysis of contemporary legal issues. I will certainly keep this in mind for future research and publications.
Three points:
1. Ensure that a non-US audience can understand the first section where you explore the case itself. Using words such as 'enjoined' will be difficult for a non-lawyer and a non-US lawyer to understand.
I have now limited, or explicitly defined, the legal jargon. Moreover, I have cleaned up language and sentence structure throughout the essay in order to improve overall readability.
2 Ensure a brief explanation of the legal provisions follow the explanation of the facts. i.e. what did the claimants and defendants have to prove on the basis of the legal provisions (explain what they say). This helps the non-specialist understand the facts and law.
Thank you especially for this remark. I have added a brief explanation of the legal provisions involved and what the applicants (for injunctive relief) had to prove in light of the provisions.
3. line 44 introduces the fact that one of the religious communities was a Jewish umbrella organisation. It would be worth introducing this earlier when you are explaining the facts. Also explaining earlier (perhaps at the same point) that you will be focussing on the RC aspects. Otherwise the non-US or non-Catholic reader might assume the judgment is about two RC churches.
I have now made clear that Agudath Israel is an Orthodox Jewish organization. This is noted in the abstract and early on in the essay.
Thank you for an excellent analysis. I really enjoyed reading this.
Thank you again for the thoughtful engagement and recommendations.
Reviewer 2 Report
An interesting analysis of religious freedom in New York during the COVID-19 pandemic. The discussion of prudence helps to place the article in a Thomistic context. However, the author draws heavily on legal writings and opinions. For the non-legal scholar, this might be a bit challenging to understand. I recognize that this paper fits into the subfield of religion and law. Not everyone may be award of the all the legal terms and context at play. The author assumes a bit.
The author also spends a bit of time discussing the value of attending Mass due the Eucharist and need to be part of the mystical body of Christ. The COVID-19 pandemic is not the first time the Church had dealt with presence of plagues. Perhaps the author can discuss how the Church has handled this in the past (e.g., during the Bubonic plaque).
Otherwise, an excellent article.
Author Response
An interesting analysis of religious freedom in New York during the COVID-19 pandemic. The discussion of prudence helps to place the article in a Thomistic context. However, the author draws heavily on legal writings and opinions. For the non-legal scholar, this might be a bit challenging to understand. I recognize that this paper fits into the subfield of religion and law. Not everyone may be award of the all the legal terms and context at play. The author assumes a bit.
Thank you for this discerning observation. In response, I have omitted unnecessary legal jargon and have defined the remaining legal terminology at their first instance. I have better explained the facts of the Brooklyn case, including in my explanation a simple explanation of the legal provisions at stake. These edits, I hope, make the legal discussion approachable for non-lawyers.
The author also spends a bit of time discussing the value of attending Mass due the Eucharist and need to be part of the mystical body of Christ. The COVID-19 pandemic is not the first time the Church had dealt with presence of plagues. Perhaps the author can discuss how the Church has handled this in the past (e.g., during the Bubonic plaque).
I have added a brief footnote, towards the end of the essay, about past ecclesial responses to plagues. Although an extended discussion of plagues may have fit within the fifth and final section of the essay (especially if such discussion illustrated the virtue of prudence), I lack the expertise and familiarity with the historical literature to do that discussion justice. That said, the gesture towards the topic, via the footnote, may pique readers' interests. I have included two solid resources on the topic in the bibliography. Thank you for this suggestion.
Otherwise, an excellent article.